# Task structure and nonlinearity jointly determine learned representational geometry

**Matteo Alleman,**[*] **Jack Lindsey**[*] **& Stefano Fusi**
Department of Neuroscience, Columbia University
`ma3811@columbia.edu, jackwlindsey@gmail.com, sf2237@columbia.edu`

## ABSTRACT

The utility of a learned neural representation depends on how well its geometry supports performance in downstream tasks. This geometry depends on the structure of the inputs, the structure of the target outputs, and the architecture of the network. By studying the learning dynamics of networks with one hidden layer, we discovered that the network's activation function has an unexpectedly strong impact on the representational geometry: Tanh networks tend to learn representations that reflect the structure of the target outputs, while ReLU networks retain more information about the structure of the raw inputs. This difference is consistently observed across a broad class of parameterized tasks in which we modulated the degree of alignment between the geometry of the task inputs and that of the task labels. We analyzed the learning dynamics in weight space and show how the differences between the networks with Tanh and ReLU nonlinearities arise from the asymmetric asymptotic behavior of ReLU, which leads feature neurons to specialize for different regions of input space. By contrast, feature neurons in Tanh networks tend to inherit the task label structure. Consequently, when the target outputs are low dimensional, Tanh networks generate neural representations that are more disentangled than those obtained with a ReLU nonlinearity. Our findings shed light on the interplay between input-output geometry, nonlinearity, and learned representations in neural networks.

## 1 INTRODUCTION

The geometric structure of representations learned by neural networks sheds light on their internal function and is key to their empirical success. The ability of networks to adapt their representations to capture the structure of training data has been shown to improve their ability to generalize (Atanasov et al., 2021; Yang & Hu, 2020; Baratin et al., 2021) and to make effective use of increasing dataset sizes (Vyas et al., 2022). Moreover, representations learned from data are essential to the success of transfer learning between tasks (Neyshabur et al., 2020).

Representational geometries dynamically evolve during network training and arise from an interaction between the structure of the inputs provided to the network, the outputs it is trained to produce, and the network architecture. In this study, we conduct an in-depth investigation of the impact of input geometry, label geometry, and nonlinearity on learned representations. We employ a parameterized family of classification tasks that allows us to probe the impact of each of these factors independently and focus on single-hidden-layer networks in which we can precisely describe representation learning dynamics over the course of training.

## 2 RELATED WORK

Prior work has observed that hidden layer representations tend to increasingly reflect the geometry of the task labels during training (Atanasov et al., 2021; Fort et al., 2020) and that this phenomenon implicitly regularizes neural network training. Theories have been developed to explain this phenomenon in linear networks (Atanasov et al., 2021; Shan & Bordelon, 2021). However, the impact

---

[*]Equal contribution

of nonlinearity on learned representational geometry is comparatively poorly understood (though see Sahs et al. (2022); Chizat & Bach (2020)). Moreover, learned network representations must extract structure besides label structure to explain the success of transfer learning (Neyshabur et al., 2020).

Many authors have studied the impact of different choices of neural network activation function. Most of the theoretical and empirical work on the subject has focused on the effect of activation function choice on network training dynamics and performance (Hayou et al., 2019; Ding et al., 2018; Ramachandran et al., 2017). By contrast, in this work, we focus on how the activation function shapes learned representations in tasks where performance is high.

Our work relates to studies on "neural collapse," (Papyan et al., 2020; Kothapalli et al., 2022), a phenomenon often observed empirically in which prolonged training causes final-layer representations in deep networks to "collapse" to represent only the label information. Prior theoretical work on the subject shows that neural collapse emerges from gradient descent dynamics in an "unconstrained features" parameterization in which the final layer network responses to each datapoint are optimized as free parameters (Zhu et al., 2021; Jiang et al., 2023). However, the parameters fit during network training are the weights of the network, and the network architecture, activation function, and input data geometry impose constraints on how network responses can be transformed across layers. Our work sheds light on how these constraints impact the propensity for neural collapse.

## 3 MEASURES OF REPRESENTATIONAL GEOMETRY

In this work, we characterize learned representational geometry in a number of ways. The tasks we consider involve mapping inputs, which are generated from a small set of binary latent variables, to one (single-output case) or several (multi-output case) binary labels.

First, we track the linear decodability (using an SVM) of different labelings of these clusters from the network representation, including those the network was trained on and those it was not. The discrepancy between decodability of trained vs. untrained labelings measures the extent to which the network preserves rich information about the inputs, or discards all but the label information.

Second, we use *kernel alignment* metrics, commonly used in assessing the similarity of two neural network representations (Cristianini et al., 2001; Kornblith et al., 2019; Kriegeskorte et al., 2008). For two mean-centered representations of a set of $d$ data points $X_1 \in \mathbb{R}^{n_1 \times d}$ and $X_2 \in \mathbb{R}^{n_2 \times d}$, we may define corresponding kernel matrices $K_1 = X_1^T X_1, K_2 = X_2^T X_2 \in \mathbb{R}^{d \times d}$. Then, the kernel alignment between these representations is defined as

$$C(K_1, K_2) = \frac{\text{Tr}(K_1 K_2)}{\sqrt{\text{Tr}(K_1 K_1) \cdot \text{Tr}(K_2 K_2)}}$$

Concretely, this corresponds to the entry-wise correlation coefficient between the kernel matrices. For inputs, $X$, output labels $Y$, and hidden representations $Z$, there are two alignment values we measure: the 'target alignment' $C(K_Z, K_Y)$, the 'input alignment' $C(K_Z, K_X)$. We also vary the 'input-output alignment' $C(K_X, K_Y)$ of our tasks.

Third, we measure the *parallelism score* (PS) of the target labels in the representation, a measure of disentanglement that indicates which a given feature is encoded in the same way regardless of the value of other input features (Bernardi et al., 2020). Concretely, in a task with $2^k$ inputs generated by $k$ underlying binary variables $y_1, ..., y_k$, the parallelism score of a representation for a variable $y_i$ is computed as follows. First, we condition on a set of values of the other $k - 1$ factors. Then, we take the vector that separates the mean value of the representation when $y_i = +1$ and the mean when $y_i = -1$. This process is repeated for all $2^{k-1}$ possible conditioned values of $y_{j \neq i}$, and the average pairwise cosine similarity between the resulting vectors is computed. Parallel coding directions of task-relevant quantities allow for better few-shot generalization and are a signature of disentangled abstract representations Higgins et al. (2018); Sorscher et al. (2021).

Finally, we measure *cross-condition generalization performance* (CCGP) (Bernardi et al., 2020), which measures the extent to which a linear classifier trained to categorize a restricted set of inputs will generalize accurately to categorizing unseen, out-of-distribution inputs. CCGP for a quantity

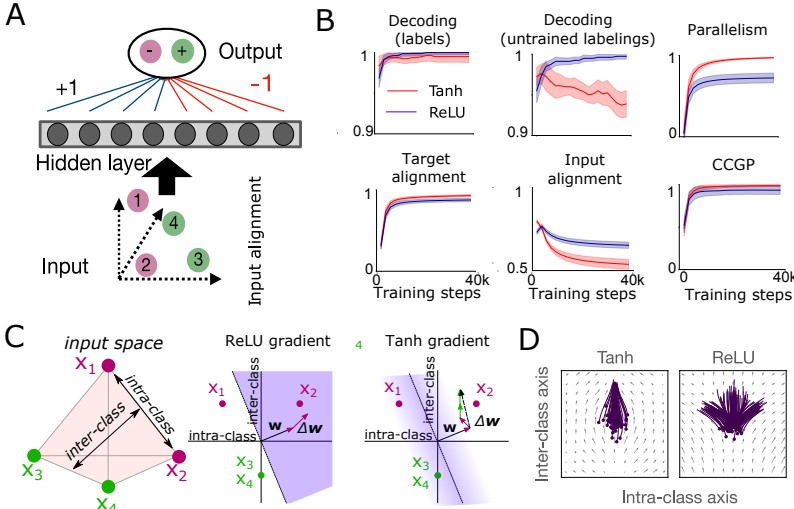

Figure 1: A. Schematic of binary classification task with unstructured inputs. B. Measures of representational geometry during training. Error bars indicate standard deviation over 20 simulated networks. C. Schematic illustrating the inter-class axis and and intra-class axis (left) and the procedure for computing the expected gradients of the task loss with respect to the input weights, projected along these axes (right two panels). The derivative $f'$ of the activation function is shown by the shading of space, and the vector $\vec{w}$ indicates the current value of the input-layer weight being considered. In this example, in the ReLU case, only the $x_2$ data point contributes to the gradient (red arrow). In the Tanh case, the gradient (dashed arrow) receives contributions from all four data points (colored arrows). D. Trajectories of input weights to hidden layer neurons along the inter-class and the intra-class axes. Each line segment represents an individual neuron from a simulation, and small circles indicate the initial conditions. Vector field indicates the gradient of the task objective.

of interest is high when the representational axes that encode that quantity are relatively unaffected by additional information about the the input, a property which enables few-shot learning Sorscher et al. (2021); Lindsey & Issa (2023); Johnston & Fusi (2023). Concretely, in a task with binary feature structure as above, to compute CCGP for a variable $y_i$, we fix a setting of all values of $y_{j \neq i}$ and fit a decoder for $y_i$ on this subsampled dataset. Then we evaluate the average performance of the decoder on all $2^{k-1} - 1$ other settings of $y_{j \neq i}$. This quantity is averaged over the $2^{k-1}$ possible choices of the setting of $y_{j \neq i}$ used for decoder training.

See Appendix D for an example that provides further intuition for the CCGP and PS metrics.

## 4 REPRESENTATIONS INDUCED BY BINARY CLASSIFICATION OF UNSTRUCTURED INPUT PATTERNS

To begin, we considered the following simple classification task. Four points $x_1, ..., x_4 \in \mathbb{R}^N$, corresponding to prototypical inputs ("cluster centers"), were sampled randomly from a unit normal distribution. Then, individual samples were drawn from normal distributions centered at the $x_i$ with variance $\sigma^2$ (set to 1.0 throughout the main text, but see Section 5.2). We trained networks with a single hidden layer to map samples from the first two clusters ($x_1$ and $x_2$) to $y = 0$ and from the last two clusters ($x_3$ and $x_4$) to $y = 1$ (Fig. 1A), with cross-entropy loss. For ease of subsequent analysis, we train only the first-layer weights and freeze the second-layer weights at binary $\pm 1$ values (all qualitative effects are robust to this choice, see Appendix B). This task is designed to assess how much the task structure (geometry of the outputs) imposes itself on the network's hidden layer representation through learning.

We found the choice of nonlinearity strongly affects the learned geometry. In particular, Tanh networks learned representations that reflected the low dimensional geometry of the targets (high target alignment, parallelism score, and CCGP), while ReLU networks learned representations that more faithfully preserved the geometry of the input clusters (high input alignment and ability to decode

untrained labelings of the input points) (Fig. 1B). Notably, in both kinds of networks, the ability to decode the class labels (the "trained dichotomy") from the network representation was high and increased throughout training. We also measured the ability to decode classes the network was not trained on. Decoding performance for such "untrained dichotomies" was high for both networks, but over the course of training, it increased in the ReLU networks and decreased in the Tanh networks.

## 4.1 ANALYSIS OF LEARNING DYNAMICS

### 4.1.1 METHODS

We next sought to understand our results by analyzing the learning dynamics of the input weights of hidden neurons. To visualize learning dynamics, we made several simplifying assumptions. As mentioned previously, we fixed the output weights of the network throughout training to discrete values, such that learning takes place only in the input weights to the hidden layer. We discretized the distribution of output weights, such that the neurons can be categorized into groups with identical output weights. Furthermore, we make an assumption about the error statistics during training, namely that task performance is approximately equal across items (e.g. the network is just as likely to correctly output the labels of input 1 as it is for input 2). Another equivalent description is that the error matrix, $Y - \hat{Y}$, has a constant direction and only changes in magnitude. This assumption is reasonable, given the symmetry of the tasks we consider.

Under these assumptions, we may describe the learning dynamics of the input weights, $\vec{w}$, of a particular hidden neuron with activation function $f$ and frozen output weights $w_o$, as follows:

$$\Delta \vec{w} = \sum_i w_o(y_i - \hat{y}_i)f'(\vec{w}^T \vec{x}_i)\vec{x}_i = \sum_i \epsilon_i f'(\vec{w}^T \vec{x}_i)\vec{x}_i \qquad (1)$$

where $i$ indexes the training examples, $\hat{y}_i$ is the output of the readout layer ( with sigmoid nonlinearity applied) for training example i, and we have grouped together $w_o(y_i - \hat{y}_i) = \epsilon_i$. Note that the evolution of $\vec{w}$ depends only on its own state, and not that of other hidden neurons. Furthermore, each hidden neuron with the same output weights (i.e. those belonging to the same "group") will be subject to the same dynamics. These simplifications allow us to generate a vector field describing network learning dynamics by plotting the $\Delta \vec{w}$ vector for an arbitrary choice of $w_o$.

We visualize the gradients in a two-dimensional space defined by the inter-class axis and an intra-class axis. The inter-class axis is equal to the covariance between input and output: $\sum_i y_i \vec{x}_i$. In the 4-input, binary classification task under consideration, it corresponds to $x_1 + x_2 - x_3 - x_4$. Intra-class axes are orthogonal to the inter-class axis, and capture differences between inputs of the same label; in this case, the intra-class axes are $x_1 - x_2$ and $x_3 - x_4$. The choice of which intra-class axis is most useful to visualize depends on the neuron being considered: for a neuron with positive output weight, the interesting dynamics take place in the $x_1 - x_2$ axis since the neuron maintains approximately zero selectivity for $x_3$ and $x_4$. Note that weights in a linear network would evolve only along the inter-class axis; dynamics along the intra-class axis are a consequence of nonlinearity. See Fig. 1C for a schematic illustrating these computations.

For a neuron with positive output weight $w_o$, the average gradient of the task loss $\mathcal{L}$ with respect to that neuron's input weights $\vec{w}$ is a sum of multiple components, each corresponding to an input cluster (Equation 1). The terms of this sum are vectors aligned with the corresponding input $\vec{x}_i$, weighted by the sign of the label of $\vec{x}_i$, and the value of $f'(\vec{w}^T \vec{x}_i)$ for the given value of $\vec{w}$ evaluated at the $\vec{x}_i$ – this weighting is indicated by the blue shading in Fig. 1C. In the ReLU case, $f'$ is 1 when $\vec{w}$ and $\vec{x}_i$ are positively aligned and zero otherwise. As a result, the gradient tends to push $\vec{w}$ further in the direction of inputs $\vec{x}$ for which it is already selective. In the Tanh case, the gradient pushes $\vec{w}$ towards inputs with the positive label (or away from inputs with the negative label) to which that neuron is neither strongly selective nor anti-selective. This has the effect of dampening strong within-class selectivity for Tanh neurons.

### 4.1.2 LEARNING DYNAMICS

As a result of the dynamics described above, in Tanh networks, all neurons grew increasingly aligned with the inter-class axis (Fig. 1D, left). ReLU networks instead exhibited heterogeneity across neu-

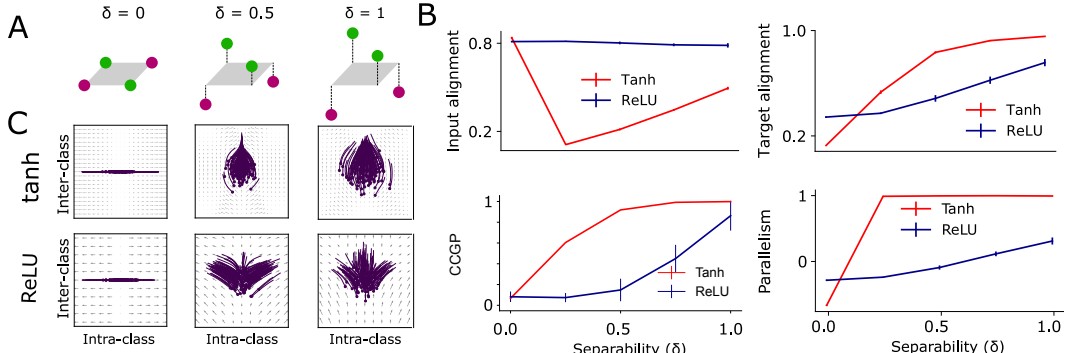

Figure 2: A. Schematic of binary classification task with input structure parameterized by $\delta$, a factor indicating the degree of separability of the two classes (green and magenta clusters). B. Measures of representational geometry following network training as a function of $\delta$. Error bars indicate standard deviation over 20 simulated networks. C. Trajectories of input weights to hidden layer neurons as in Fig. 1D, for different values of $\delta$.

rons, with some growing aligned with the inter-class axis and others developing intra-class selectivity (Fig. 1D, right). To understand this difference, we looked at the dynamics of gradient descent. In Fig. 1D we plot the vector field defined by projecting the expected gradient of the task objective with respect to input weights onto the inter- and intra-class axes, as described above. We find that weights in Tanh networks predominantly evolve in a direction of increased inter-class selectivity and decreased intra-class selectivity, independent of their initial conditions. Weights in ReLU networks are driven to accentuate the selectivity they possess in their random initial condition.

The different dynamics are explained by the degree of symmetry of the nonlinearity of the activation function: the one-sided saturating behavior of ReLU causes some inputs to be encoded in the weights (those that bring the neuron above threshold) and others to be ignored (below threshold). This breaks the symmetry between the input items that would share the same gradient in the absence of the nonlinearity, leading to specialization and elevated intra-class selectivity. This symmetry breaking is less likely in Tanh neurons, given that Tanh is symmetric (both around the origin, which governs dynamics at initialization, and asymptotically). We explore the relative importance of the behavior of the nonlinearities at initialization vs. their asymptotic behavior in Section 9.

## 5 EFFECT OF INPUT GEOMETRY ON LEARNED REPRESENTATIONS

### 5.1 EFFECT OF SEPARABILITY OF TARGET OUTPUTS

We generalized the previous task by parametrically controlling the geometric arrangement of the cluster centers $x_1, ..., x_4$, rather than sampling them randomly. Specifically, the input geometry was parameterized by a scalar quantity $\delta$ corresponding to the degree to which the two classes were linearly separable in the input space. The $\delta = 1$ case corresponds to the previous task, in which the clusters are equidistant, and hence the trained dichotomy is easily decodable (Fig. 2A, right). In the $\delta = 0$ case, the clusters were arranged on a two-dimensional square such that $x_1$ and $x_2$ (and $x_3$ and $x_4$) were positioned on opposite corners of the square (Fig. 2A, left), which is an XOR task and requires a nonlinear transformation of the inputs. Intermediate values of $\delta$ interpolated between these two extremes (Fig. 2A, middle). Note that this construction can be regarded as varying the input-output kernel alignment of the task (low $\delta$ corresponds to lower alignment).

In the full XOR task ($\delta = 0$), no matter the nonlinearity used by the network, a representation emerged in the hidden layer with low target alignment, parallelism score, and CCGP (Fig. 2B). This finding is unsurprising, as in this task, a disentangled representation of the output geometry cannot be obtained by one neural network layer. For intermediate values of $\delta$, however, it is not obvious to what extent the network will leverage the linearly separable component of the inputs for disentanglement. We found that Tanh networks, for any values of $\delta$ greater than a small threshold, form representation in which the geometric structure of the inputs was largely discarded in favor of the binary output structure (Fig. 2B). For ReLU networks, the alignment of the representation with

the output structure increased more gradually as $\delta$ varied from 0 to 1, and strong signatures of the input structure remained in the learned representation regardless of the value of $\delta$.

To understand these effects, we again analyzed the evolution of the input weights with learning, focusing as before on the inter-class axis and intra-class axes. We found that for $\delta = 0$, only intra-class selectivity emerged for all neurons in all networks, which is unsurprising as the inter-class axis in this case is degenerate $((x_1 + x_2) - (x_3 + x_4) = 0)$. For any nonzero values of $\delta$, Tanh neurons almost uniformly developed inter-class selectivity, while ReLU neurons evolved in a heterogeneous fashion, with the proportion of intra-class neurons decreasing with $\delta$ (Fig 2C).

## 5.2 Effect of input noise

We also assessed the impact of input noise on representation learning (see Fig. 3). To facilitate a fair comparison of learned representations across networks, we introduced a distinction between degree of input noise $\sigma_{train}$ and $\sigma_{test}$ used during training and during analysis of learned representations, respectively. The value of $\sigma_{test}$ was fixed at 1.0 for all analyses, and the value of $\sigma_{train}$ varied. We found that Tanh networks exhibit a sharp transition in learned representation, from abstract representations to input geometry-preserving representations, as training noise increases. The level of noise at which this transition occurs is related to the separability of the trained dichotomy; for high values of separability, learned abstraction is robust to higher levels of noise during training.

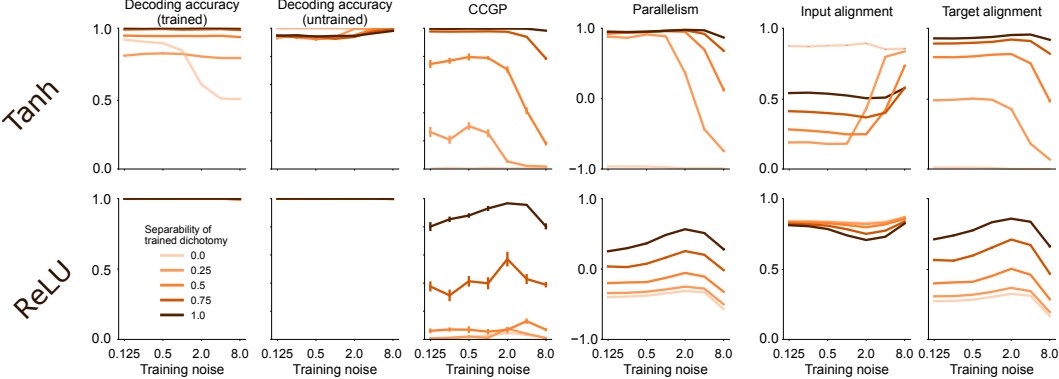

Figure 3: Values of various representational metrics for different values of $\delta$ (separability of trained dichotomy) and $\sigma^2$ (training noise) in the $\delta$-separable classification task of Section 5.

The effect of increased noise in Tanh networks is to induce more "ReLU-like" representations that vary gradually as a function of the input geometry. By comparison, learned representations in ReLU networks are impacted much less by training noise. This behavior makes sense, given the analysis in the previous section revealing that Tanh networks are dramatically affected by the separability of the target outputs in input space, while ReLU networks are less sensitive to the degree of separability. Increasing the degree of input noise has the effect of increasing the degree of separability required for Tanh neurons to reliably tune to the component of input space that correlates with the label. When separability is insufficiently strong given the noise level, Tanh network behavior grows more similar similar to the $\delta = 0$ case. Interestingly, we also observe an increase in target alignment and a decrease in input alignment for modest values of input noise in ReLU networks. We explore this phenomenon further in Appendix E.

## 6 Generalizing analysis to more complex tasks

So far, we studied in depth a set of tasks involving just four input clusters. Now we check if the insights derived from these simple tasks hold more broadly. In the $\delta$-separable XOR task, increasing the separability parameter makes the input geometry more similar to the output geometry, resulting in a higher target alignment in the hidden layer. We generalize this construction to assess more generally how the kernel alignment between input and target patterns determines hidden layer geometry. To do so, we use the following procedure to sample tasks with a specified input-output alignment.

We parameterize a family of tasks with $P$ randomly placed input clusters and $k < P$ binary output targets, in which we control the alignment between input and target kernels (Fig 4A). To do so, we first draw $k$ random but balanced binary target classes, $Y$. Then, for a specified alignment value $c$, we draw a random input kernel, $K_X$, from the set of all symmetric positive definite matrices such that $C(K_X, K_Y) = c$, where $K_Y = Y^T Y$. One element of this set is special, and it is the $K_X$ with the flattest eigenspectrum, i.e. the maximal linear dimensionality. The solid lines in Fig. 4 use this maximal-dimensional input geometry, while the dots use other random draws with lower dimensionality. With $K_X$ in hand, there are many ways we can generate $N$-dimensional input patterns $X$, and we use $X = O\Lambda^{\frac{1}{2}}U^T$, where $U$ and $\Lambda$ are the eigenvectors and eiegenvalues, respectively, of $K_X$, and $O$ is a random $N \times P$ orthonormal matrix. We end up with a set of random inputs $X$, and output labels $Y$, with a centered kernel alignment of exactly $c$.

In Fig 4B, we plot our measures of representational geometry for $P = 8$ and 32 inputs, varying $k$, the number of targets, from 1 to $P - 1$. As in the simple case of Section 5, the target alignment always increases more dramatically for Tanh networks compared to ReLU as input-output alignment increases. Moreover, as expected, the parallelism score increases when the target geometry is low-dimensional and decreases when it is high-dimensional, and hence the target geometry itself has low parallelism. Note that the effect of multi-dimensional outputs is explored in more depth in a case study ($k = 2$) in Appendix A. The behavior of CCGP mostly matches that of parallelism, but there are several cases where a large difference in parallelism is not reflected by a large difference in CCGP, particularly in cases with relatively few outputs, where the decoding task measured by CCGP may be easier.

## 7 PHENOMENA IN MULTI-LAYER NETWORKS

We wished to see whether our findings on simulated tasks generalize at all to networks with more than one layer. To address this, we chose one of the tasks from Figure 4 (that with $P = 32$ inputs and $k = 5$ targets) and trained networks with 5 or 10 hidden layers. We focused on tasks in which the outputs were nonlinearly separable in input space, as real-world tasks require dealing with nonlinearly separable inputs, and such tasks are most likely to expose differences between deep and shallow networks (target-aligned representations cannot be produced in a single-layer for nonlinearly separable targets). We evaluate on two tasks, varying in their difficulty; intuitively, the difficulty is the degree to which the output labels are nonlinearly entangled in the input space (see Appendix F for a more precise explanation). The target and input alignment evolve as expected along the layers of the network, with target alignment increasing and input alignment decreasing (Fig. 5). The effects of non-linearity and task difficulty are also consistent with our results from shallow networks – deep tanh networks learn more target-aligned representations than deep relu networks in their final layer, especially for the more difficult task. We also observe interesting phenomena with respect to the progression of target / input alignment across layers of the network in the easy vs. hard task, which we leave to future work to investigate in more detail.

## 8 CONVOLUTIONAL NETWORK EXPERIMENTS

To assess the applicability of our findings to more realistic tasks, we trained convolutional networks image classification task, experimenting with two architectures – a small network with two convolutional and two fully connected layers, and the ResNet-18 architecture – and two datasets, CIFAR-10 and STL-10. To enable computation of CCGP and parallelism score, we modified the tasks slightly so that the 10 classes in the dataset were assigned to 5 labels, grouping together pairs of classes. This allowed us to treat the input classes analogously to the input clusters in the simulations above for the purpose of computing CCGP and PS. Kernel alignment and test accuracy were also computed. In all cases, we observe the same qualitative dependence of all these metrics on the choice of activation function (though the effects vary in magnitude) (Fig 4C).

## 9 ISOLATING THE ROLE OF ACTIVATION FUNCTION ASYMMETRY

Finally, we sought to identify the source of the different representations learned by Tanh and ReLU networks. We hypothesize two candidate mechanisms: the symmetric saturation of the Tanh func-

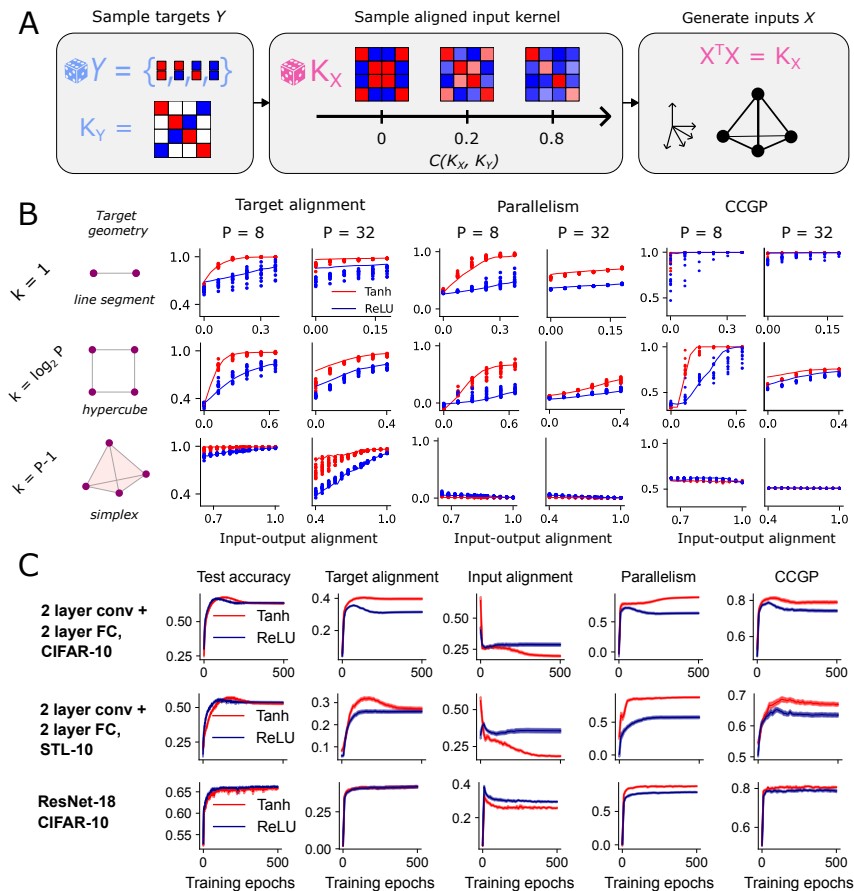

Figure 4: A. Cartoon of the random sampling process, illustrated for $P = 4$ inputs and $k = 2$ outputs. B. Target alignment, PS, and CCGP as functions of input-output alignment in random classification tasks, for different values of $P$ (columns) and $k$ (rows). Cartoons schematize the target geometry for each value of $k$ (the number of target dimensions). In the plots, solid lines are the unique maximum-dimensional input geometry for specified alignment, and dots are 12 random samples of other lower-dimensional geometries. All tasks have a training noise variance of 1. C. Metrics in the final layer of a convolutional network trained on CIFAR10. Error bars are standard errors over random initializations.

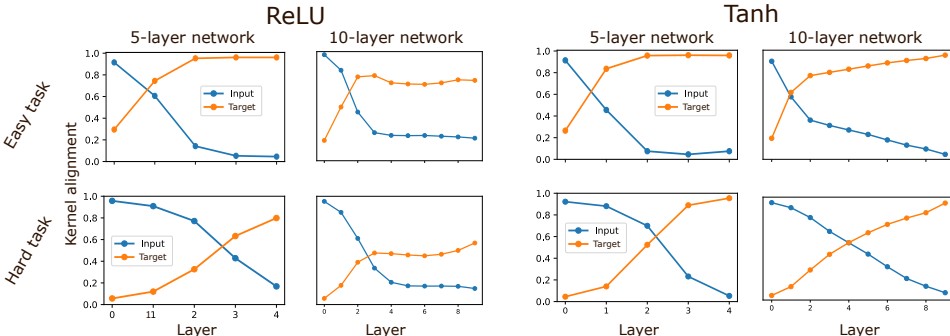

Figure 5: Target and input kernel alignment for the representations at each hidden layer in multi-layer networks. Each network is trained until convergence. The inputs are generated as in Fig. 4, with the addition of a constraint on the dimensionality of the inputs for the 'hard' task (see Section 7 and Appendix F). All tasks have a training noise variance of 1.

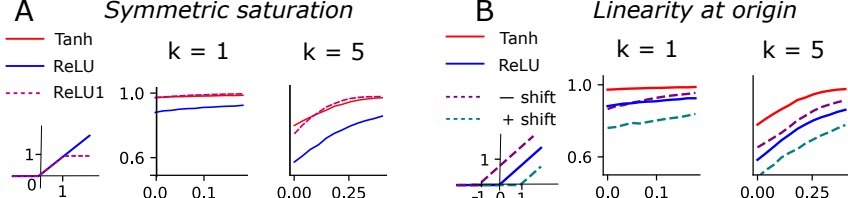

Figure 6: Activation function perturbation experiment. The target alignment is shown, as in Fig. 4, for $P = 32$ inputs. A. Adding positive saturation to the ReLU function for arguments $> 1$. B. Shifting the ReLU function negatively (reddish) or positively (greenish) in the argument.

tion, and its linear behavior close to the origin. Our analysis of the gradients in the simple task of Section 4 suggests that the degree of symmetry in the asymptotic behavior of the nonlinearity is important since it prevents individual neurons from developing selectivity for one input over another. However, it is also plausible that the behavior of the nonlinearity locally around the origin matters most if networks learn solutions that do not deviate much from their initialization. To differentiate between these hypotheses, we construct nonlinearities with symmetric asymptotic saturating behavior but asymmetric behavior around the origin (Fig. 6a, f(x) = min(max(x, 0), 1)), or asymmetric saturating behavior but linear behavior around the origin (Fig. 6b, f(x) = max(x+b, 0), b<0).

We find that networks using a nonlinearity with two-sided saturation (Fig. 6a) behave almost identically to Tanh networks despite asymmetry around the origin. Recall from Fig. 1C, D that in the Tanh case, the two-sided saturation of the nonlinearity prevents neurons from growing overly selective or anti-selective for particular inputs with a given label, as when this occurs, the value of $f'$ evaluated at those inputs is low, dampening the magnitude further weight updates aligned with that input direction. Qualitatively, the same phenomenon occurs in the gradients of any nonlinearity that saturates in both the positive and negative directions.

We also tried perturbing the offset of ReLU to make it linear and symmetric around the origin without modifying its asymptotic behavior. We found this has a modest effect on learned representational geometry, leading to more target-aligned representations when the linear region of the ReLU nonlinearity contained the origin (Fig. 6b). This makes sense, as linear or otherwise symmetric behavior of the activation function derivative $f'$ around the initialized value of the input weights should result in an initial evolution of input weights along the inter-class axis with learning, until they reach a region of weight space where $f'$ is asymmetric with respect to the inputs $\vec{x}$.

We conclude that activation function behavior around the origin influences learned solutions, but symmetric asymptotic saturating behavior exerts a powerful influence towards target alignment.

## 10  CONCLUSIONS AND DISCUSSION

The geometry of learned neural representations combines the structure present in inputs and target outputs, influencing a network's task performance and ability to generalize to new data. Here, we introduced a framework for modeling the joint structure of task inputs and outputs, and we studied how neural representations reflect these structures. Surprisingly, the activation function plays an important role in determining the alignment between the learned representational geometry and the target geometry, with Tanh typically leading to more disentangled representations of the structure of the labels than ReLU. These differences in learned representations trade off different benefits. Disentangled representations are compact Ma et al. (2022), allow for generalization and compositionality, and have been shown to improve adversarial robustness (Willetts et al., 2019; Yang & Hu, 2020; Papyan et al., 2020). However, they are inefficient representations for storing memories and for binding together information about multiple variables Boyle et al. (2022); Johnston et al. (2023). Moreover, learning disentangled representations of the label structure may impair the ability to transfer learning to tasks with different semantics. Our work sheds light on the aspects of network architecture and task structure factors that are important in navigating this tradeoff.

## 11 Acknoledgements

This work was supported by NSF NeuroNex Award DBI–1707398, The Simons Foundation, The Gatsby Foundation (GAT3708), the Swartz Foundation and the Kavli Foundation. JL was also supported by the DOE CSGF (DE–SC0020347). The authors declare no competing interests.

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

## A    EFFECT OF MULTIDIMENSIONAL OUTPUT ON LEARNED REPRESENTATIONS

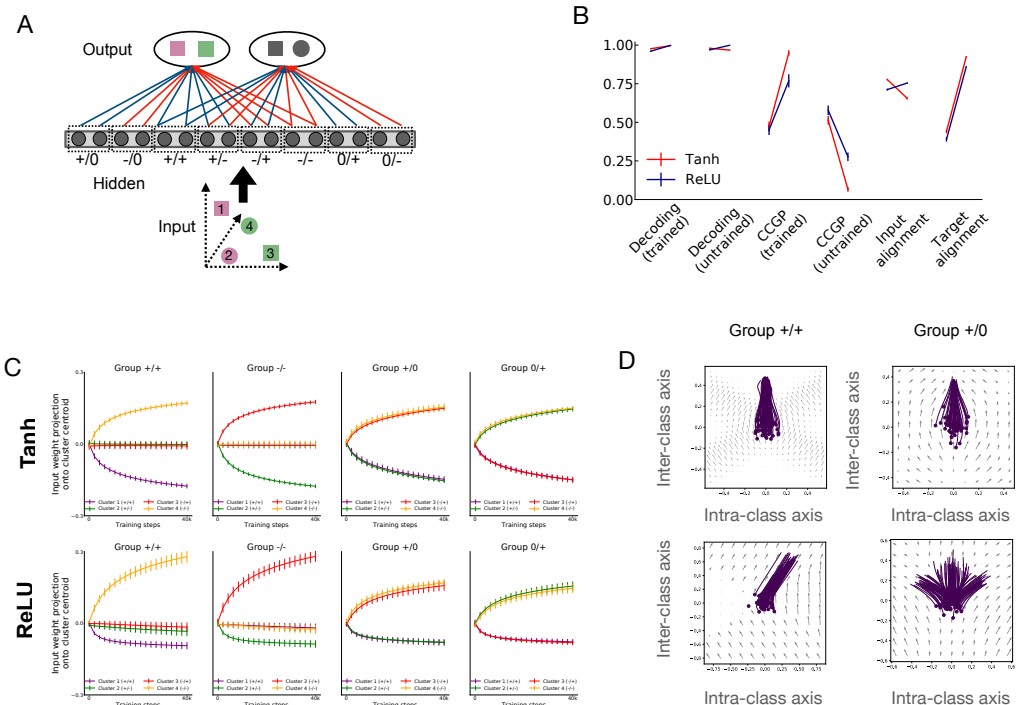

Figure 7: A. Schematic of binary classification task with unstructured inputs and two outputs. The outputs are trained to classify four input clusters according to two different labelings of the inputs. In this case the hidden layer of the network is divided into eight groups of neurons defined by their two output weights (positive: blue, negative: red, zero: no line). B. Measures of representational geometry at network initialization (left of each line segment) and following training (right of each line segment). Error bars indicate standard deviation over 20 simulated networks. D. Alignment (dot product similarity) of input weights to hidden layer neurons with the four input cluster centers. Shown for four of the eight hidden layer neuron groups. Error bars indicate standard deviation across neurons within a single network. E. Trajectories of input weights to hidden layer neurons along the inter-class ($x_1 + x2 - x_3 - x_4$) and the intra-class axis ($x_1 - x_2$). Each line sigment represents an individual neuron from a simulation, and small circles indicate the beginning of the trajectory (at network initialization). Shown for hidden layer neurons with an output weight of $+1$. Vector field background indicates the expected average direction that weights will evolve due to gradient of the task objective.

Here we include an in-depth exploration of networks trained with two targets and four items. We modified our tasks by now assigning each of the four input clusters to two different binary labels: $(1, 1)$, $(1, 0)$, $(0, 1)$, and $(0, 0)$, respectively (Fig 7). We modified the network architecture by adding a second output unit; each output unit was tasked with predicting one of the two labels. Thus, in this task, there were two trained dichotomies ($x_1/x_2$ vs. $x_3/x_4$, and $x_1/x_3$ vs. $x_1/x_4$) and one untrained dichotomy ($x_1/x_4$ vs. $x_2/x_3$). For analysis, we again used frozen readout weights discretized into eight groups (Fig 7A), which we found was sufficient to approximate the behavior of networks trained with randomly initialized output weights. Again, we found a noticeable difference in the learned representations of ReLU and Tanh networks, with Tanh networks reflecting the geometry of the targets more than ReLU networks (Fig 7B). Again, we analyzed the source of these phenomena by tracking the alignment of input weights to each neuron with the four input cluster centroids. In general, we observed that neurons grew positively tuned to inputs which matched their

output weights, negatively tuned to inputs mismatched with their output weights, and not tuned at all to inputs matched with one output weight and mismatched with the other (Fig. 7C). However, in the ReLU networks, a clear asymmetry emerged in the degree of positive tuning to matched input clusters and the degree of negative tuning to mismatched input clusters. As a result, in the ReLU networks, roughly half the neurons (those with nonzero outputs to both class readout units) developed selectivity primarily for one of the four input clusters. As a result, these ReLU units developed strong selectivity for intra-class differences, while Tanh units did not (Fig. 7D). This behavior was observed to be uniform across the hidden neurons (Fig. 7D).

Thus, both the multi-output task produced similar results as the similar output task in terms of representational geometry, the source of these effects was somewhat different. In both cases, the gradient of the task objective drives input weights to accrue inter-class selectivity in Tanh networks. However, for ReLU networks, the picture is different. In the single-output case, the emergence of intra-class selectivity arises due to heterogeneity in initial conditions of input weights, while in the multi-output case it also arises due to heterogeneity in initial conditions of output weights.

## B  IMPACT OF CONSTRAINTS ON THE OUTPUT WEIGHTS

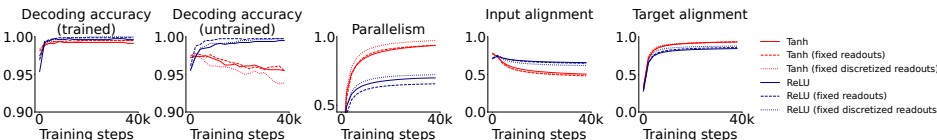

Figure 8: Measures of representational geometry during training for three cases: randomly initialized trainable readouts, randomly intiialized fixed readouts, and discretized (binary $\pm 1$) fixed readouts.

In our experiments we used frozen output weights with discretized values. In Fig. 8 we show that the behavior of representation learning is essentially the same when we relax the discretization and allow the readouts to be trained.

## C  FURTHER DETAILS ON RANDOM ALIGNED KERNEL SAMPLING

Our sampling procedure for tasks with a specified input-output alignment used in Section 5 has some important considerations, which will be further elaborated in this section. Firstly, obviously, the sampled matrices must be symmetric and positive semi-definite (SPSD), and they need to have the required correlation value. Secondly, kernels with the same alignment value can have different dimensionality (i.e. eigenspectra).

It is worth clarifying that we assume all kernels are already centered, meaning that the features used to compute them have been mean-subtracted. Or, equivalently, that their nullspace contains the vector $\mathbf{1}$ of all ones. They should also have unit trace, $\text{Tr} K = 1$. Both are non-restrictive assumptions, since the CKA is invariant to scaling and shifting of features.

The set of matrices with a given correlation value, $c$, to a reference matrix $K_Y$ is the solution set of a quadratic form[1], while the SPSD matrices (of unit trace) are a compact convex body. We will refer to this set as $\mathcal{K}_c$. Our procedure is to randomly sample a SPSD matrix, and project it onto the quadric surface, taking care to remain SPSD. This method certainly does not ensure uniformity, but it is simple and empirically shows a wide spread of samples even in relatively high dimensions.

An important consideration with this method is the linear dimensionality of the input geometry, intuitively a measure of correlations between input patterns. If we define dimensionality in terms of the participation ratio of the eigenvalues, $\lambda$, of $K$:

$$\text{p.r.}(K) = \frac{\left(\sum_i \lambda_i\right)^2}{\sum_i \lambda_i^2}$$

---

[1] $C(K_X, K_Y) = c \Leftrightarrow \text{Tr}(K_X K_Y)^2 = c^2 \text{Tr}(K_X K_X) \text{Tr}(K_Y K_Y)$.

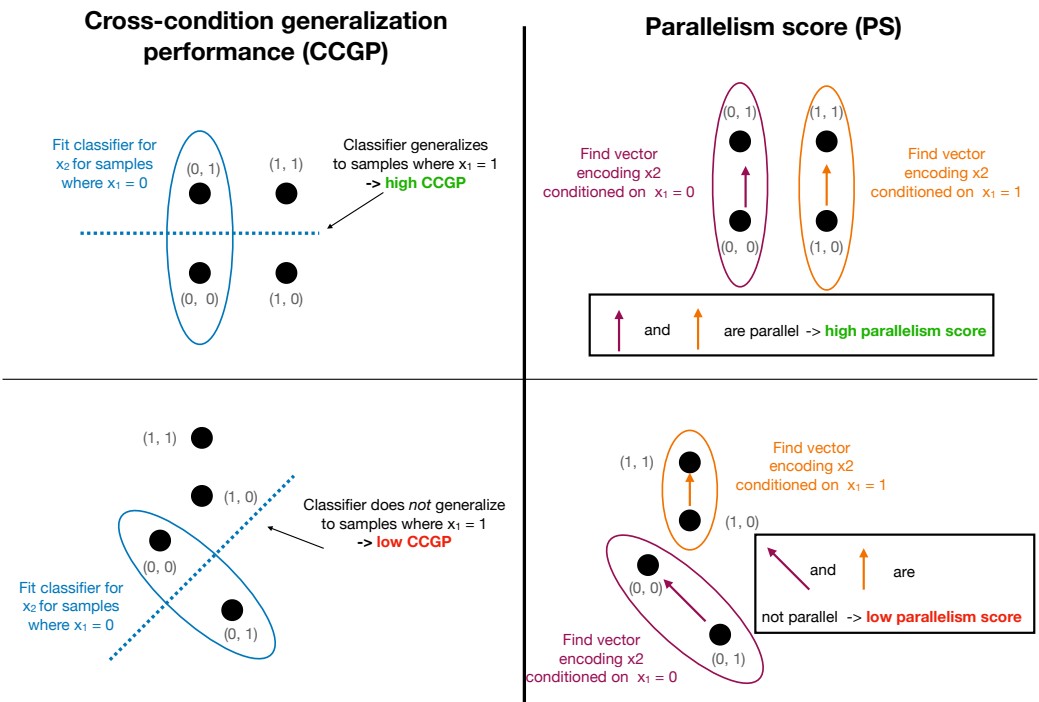

Figure 9: Examples of representational geometries with high and low CCGP and PS, for data generated by two underlying binary variables, i.e. data points of the form $(x_1, x_2)$ where $x_i \in \{0, 1\}$.

then, per our assumption of unit trace, this is just $\|K\|_F^{-2}$. In the special case when $c = 0$ the set $\mathcal{K}_0$ is convex, and this implies that there is a unique kernel matrix $K_{\max}$ which maximises $\mathrm{p.r.}(K)$, i.e. minimises $\|K\|_F$. If we draw a line between $K_Y$ and $K_{\max}$, we get the set of input kernels which form the solid lines of Fig. 4B and Fig. 5 of the main text.

## D  EXPLANATION OF CCGP AND PS METRICS

See Figure 9.

## E  COMPARISON OF WEIGHT DYNAMICS AT DIFFERENT NOISE LEVELS IN RELU NETWORKS

In Section 5.2 we observed in ReLU networks an interesting increase in the values of target alignment / parallelism / CCGP, and decrease in the value of input alignnment, for intermediate values of the input noise $\sigma$. Here (Fig. 10) we provide an analysis that sheds some light on this phenomenon. For different noise values, we tracked the relationship between the value of the input weights to a network hidden-layer neuron and the corresponding value following training. In the absence of noise, neurons initialized with sufficient intra-class selectivity relative to their initial inter-class selectivity are destined to maintain it over time (remain in the purple or green regions in Fig. 10 – see weight trajectory plots in Fig. 1D, 2C for an explanation of why). In the presence of noise, neurons with substantial intra-class vs. inter-class selectivity at initialization can nevertheless evolve to develop primarily inter-class selectivity over the course of training. Note that this explanation does not account for why, at extremely high noise values, target alignment drops again (Fig. 3C) – we leave an in-depth characterization of this phenomenon to future work, but suspect it arises because the input noise grows substantial enough for the weights to accrue significant projections along axes other than the two we visualize here.

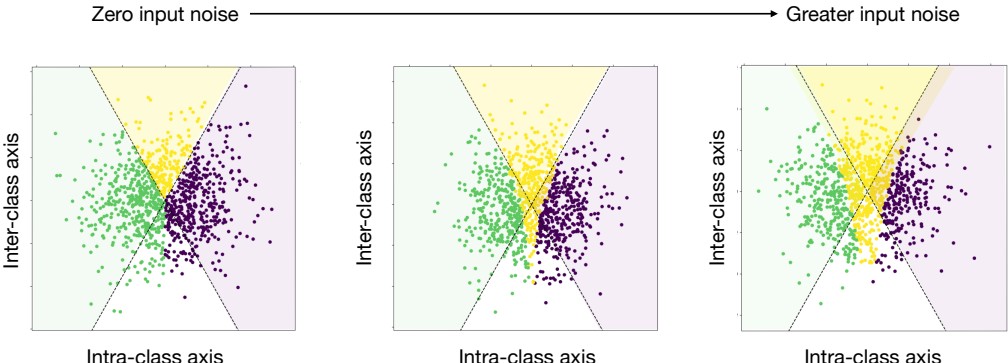

Figure 10: Effect of input noise on training dynamics of ReLU networks trained on the tasks in Section 5.2, in the case of maximal separability of the trained dichotomy, varying the level of input noise $\sigma^2$ during training. Each point corresponds a hidden-layer neuron in the network, and the position of the point indicates the value of the input weights for that neuron at initialization. The colors of the dots indicate which of the three colored regions the input weights end up in at the end of training.

## F   EXPLANATION OF TASKS USED IN SYNTHETIC MULTI-LAYER NETWORK EXPERIMENTS

Here we describe the tasks used in Section 7. Formally, the "hard" task is generated by sampling input data of minimal possible dimensionality (5), or equivalently an input data kernel of rank 5, subject to the constraint of zero input-output kernel alignment. Intuitively, this task involves inputs generated by 5 binary latent variables, and the input-output function requires a nonlinear conjunction of all 5 (e.g. whether the number of latent variables set to 1 is even or odd).

The "easy" task involves sampling inputs of maximal possible dimensionality (27 dimensions, i.e. a rank-27 input data kernel) subject to the kernel alignment constraint. Intuitively, this task, the targets require nonlinear conjunction of only two of the input dimensions.