# OpenReview forum: "Task structure and nonlinearity jointly determine learned representational geometry"
_ICLR.cc/2024/Conference — ICLR 2024 poster_

### Official Review · Reviewer_dzyA · 2023-10-31

**Soundness:** 3 good
**Presentation:** 2 fair
**Contribution:** 2 fair
**Rating:** 5
**Confidence:** 2

**Summary:**

The paper studies the learning dynamics of 1-hidden-layer neural networks, and focuses on how representation properties are affected by the nonlinearity of the network. Specifically, the paper compares Tanh and ReLU nonlinearity, and show that Tanh networks tend to reflect structure of target outputs while ReLU networks tend to retain more information about the inputs.

**Strengths:**

The paper presents an interesting empirical study on how nonlinear activation functions affect the learned representation in the network, specifically how they align with the target output and the input. The results are evaluated using several previously established metrics, starting from simple toy model dataset, the authors carefully analyzed the effect of input geometry on learned representations, and later extended their results to more complicated tasks.

**Weaknesses:**

The paper largely focuses on ReLU and Tanh nonlinearity, which are two very specific type of activation functions. It would be nice if the authors can identify what exactly is the property of the nonlinear function that causes the difference in representation, and evaluate further (as they show in Fig. 5 but with more extensive results).
The paper also presents mostly empirical evaluations and analysis without theoretical insights.

**Questions:**

In most of the work the result focuses on 1-hidden-layer neural networks but in the convolution part there are two FC hidden layers. I'm wondering how depth affect your current observations.

---

> ### Author Response · Authors · 2023-11-19
> **Response to reviewer dzyA**
>
> We thank the reviewer for the helpful suggestions and questions.
>
> – We have added substantial additional explanation in Fig. 4.1.1 explaining our construction of the vector field plots in Fig. 1D and 2C, which we regard as our main theoretical insight, and providing further intuition for how activation function differences give rise to corresponding differences in gradient dynamics in weight space.  We have also added substantial additional explanation to Section 9 (formerly Section 7), which pinpoints relative contributions of different aspects of the activation function (local behavior around the origin, and asymptotic behavior) that results in emergent representational differences.
>
> – Regarding exploring the role of depth: see our “response to all reviewers” for an explanation of new experiments we have added to the paper with deep convolutional networks of different architectures, and deep multi-layer perceptrons trained on synthetic tasks.

---

### Official Review · Reviewer_HXSW · 2023-11-01

**Soundness:** 3 good
**Presentation:** 3 good
**Contribution:** 2 fair
**Rating:** 6
**Confidence:** 3

**Summary:**

The authors systematically conducted a series of gradually more intricate experiment to investigate how nonlinearity, label separability, and input data geometry affects the learned representation in the hidden layer of 2 layer MLP, with potential generalization to CNNs. They quantified representation geometry with many statistics, including the CKA with input data, output label; parallelism index; CCGP; classifying ability for unseen label. They found a central unexpected theme that Tanh network seems to keep more geometry about the target label; while relu network seems to align the representation with the input data, and keep more target-unrelated information about the input. Finally, they dissect the source of differences between the nonlinearities ReLU and Tanh, and found that the double sided symmetric saturation of Tanh function probably explain their difference.

**Strengths:**

- Clear results. Though the general theme is expected, the relevance of activation function still has a little bit surprise.
- The experiments are systematic and well-designed, which formed an investigation with clear logic. The statistics for quantifying the representation geometry are quite comprehensive and laudable.
- The way of visualizing the training dynamics in weight space along intra-class vs inter-class axes is illuminating, further this analysis of toy model indeed provides intuition for the phenomena regarding different gradient learning dynamics for different activation functions (at lease in 2-layer networks).
- Nice controlled experiments to parse out factors explaining difference between relu and tanh, in Sec. 7, showing that the rough two-sided saturation shape is the key. in another perspective, the banded gradient structure of activation function is key.
- The claim, (if it’s general) will definitely impact how we understand the representation similarity between two systems, e.g. the brain vs CNNs. Namely, if neurons in the visual brain are using a different activation function from the CNNs, even the underlying linear function is the same, the similarity matrix won’t match.

**Weaknesses:**

- Most of the experiment focused on toy scale examples of binary classification with 2-layer network, even experiments with CNN has only 2 conv-layer networks. I feel it’s within the scope of this paper to show empirical evidence that the observations may generalize to larger scale CNN and larger dataset. (e.g. resnet and ImageNet) Will it be feasible to swap the activation function and show some similar effects?
- Notation in Eq. (in Sec. 3.1.1), or the method description in Sec. 3.1.1 is a bit confusing. Is it using multi-output setup so $W_O$ is also a vector?
- There seems to be super interesting intuition going on in Figure 1C, but the text in Sec. 3.1.1 didn’t seem to walk through the logic, leaving the reader to parse the schematics themselves. —— though after working through the math of an example it starts making sense.

**Questions:**

- The assumptions for making the weight learning animation seems a bit strong, can we alleviate those assumptions? (fixed output weights and discretized output weights)?
- How to generalize the learning dynamic visualization to non-two-layer deeper networks?
- Is there a typo of formula in Sec 3.1.1 “*the covariance between input and output: xxx*" should it be $y_i$?

---

> ### Author Response · Authors · 2023-11-19
> **Response to reviewer HXSW**
>
> We thank the reviewer for the thorough reading and insightful comments. We have addressed the questions/concerns brought up by the reviewer:
>
> – Regarding the concerns about our focus on simple networks and tasks, see our “response to all reviewers” and Section 8 in the revised manuscript for a description additional results with deep convolutional networks on realistic tasks
>
> – We agree the w_o vector notation was confusing.  In principle, w_o and the labels y can be multi-dimensional vectors (as in our Fig. 4 experiments).  However, our experiments in Section 4 focus on the single-output case, so we have changed the notation to defining w_o as a single scalar for each hidden-layer neuron to avoid confusion.
>
> – Regarding the explanation of Figure 1C, we agree that the explanation in our initial submission was inadequate, and so we have added a paragraph to accompany the schematics (in red text in section 4.1.1 in the revised paper).
>
> – The assumptions for the vector field plots in Fig. 1D and 2C, about discretized and fixed out weights and balanced errors, are indeed strong.  However, in Appendix C / Fig. 2 of the supplementary material we show that the trajectories of representational metrics for networks trained in the standard fashion with standard weight initializations agree closely with the predictions of the simplified model.
>
> – Thank you for pointing out the typo in section 3.1.1 (now section 4.1.1 in the revised manuscript), we have fixed it to say y_i.
>
> – Regarding how to generalize the visualizations to deeper networks: this is a great question, and one we are interested in exploring in follow-up work!  We expect that linearizing the network around its current activation values to plot the “effective” weights from the input to deep-layer neurons may be a useful approach.

---

### Official Review · Reviewer_eLPi · 2023-11-08

**Soundness:** 3 good
**Presentation:** 3 good
**Contribution:** 3 good
**Rating:** 8
**Confidence:** 3

**Summary:**

The current paper discussed how ReLu and Tanh activation functions impacted the representation geometry of a single layer feedforward neural network. The authors found that Tanh nonlinearity tend to generate target aligned representation, while RuLe nonlinearity favors input aligned representation. It seems the symmetric saturation of the nonlinearity is the key for target-aligned representation of the Tanh function.

**Strengths:**

The paper has thoroughly studies the representation geometry using various geometry matrix, which allowed authors to generate insights on whether a network generate input- or target- aligned representation in the hidden layer of a single layer feedforward network. It shows how the representation geometry evolves over the course of learning. In particular the trajectories of input weights to hidden layer neurons for inter-class and intra-class labels is interesting.

**Weaknesses:**

In general, the results generated by the current study that Tanh nonlinearity helps generate target aligned representation is limited to simple networks and simple input-output mapping. The representation geometry in these simple networks probably are not sufficient for many real-world problems that require capturing intricate patterns.

**Questions:**

1.	Why the decoding accuracy is worse in training data with high separability of trained dichotomy for Tanh network (figure 3 upper left)? The separability has non-monotonic effects on input alignment in the Tanh network, why is that?
2.	Why the noise level has non-monotonic effects on the Relu network, consistently observed in all geometric matrix and for all tested separability of trained dichotomy? Author suggested smoothing gradients. What’s the evidence supporting such conclusion?
3.	Are the results robust to the input data range, eg. Input data ranges between 0 to 1, vs. input data ranges between -1 to 1.
4.	When training data becomes more complicated, as suggested by the noise input analysis and the XOR task, the difference between tanh and ReLu vanish. This suggested that Tanh helps generate a target-aligned representation when the input data is readily separated for clustering. Or in a multi-layer network, using tanh at the final layer seems to be beneficial. This is not a novel conclusion. What’s the new insights learnt from the current study. Provide a discussion on how the results learnt in the current study would have a general impact.

---

> ### Author Response · Authors · 2023-11-19
> **Response to reviewer eLPi**
>
> We thank the reviewer for the positive comments about the thoroughness of our analysis and the insights gained.  We have addressed the questions/concerns brought up by the reviewer:
>
> – Regarding the concerns about our focus on simple networks and tasks, see our “response to all reviewers” and Section 8 in the revised manuscript for a description additional results with deep convolutional networks on realistic tasks
>
> – Regarding the apparent strange result of lower decoding accuracy in more highly separable tasks in figure 3: this was a mistake on our part; we mistakenly inverted the color scale in that panel of figure 3 in our initial submission!  Please see the revised manuscript for the corrected figure, and thank you for bringing the error to our attention
>
> – Regarding the effects of noise on learned representations in ReLU networks (its non-monotonic effect on target alignment and other representational metrics): we have conducted an additional analysis investigating this phenomenon in Appendix E in the revised manuscript.  In short, increased training noise makes it more likely for units that are initialized with high intra-class selectivity to “escape” regions of weight space with high intra-class / low inter-class selectivity, increasing target alignment.  We believe this phenomenon explains why intermediate levels of noise increase target alignment relative to zero noise.  However, we do not yet have a full mechanistic explanation for the non-monotonicity (i.e. why very high noise levels reduce target alignment); we suspect it arises due to noisy weight update dynamics outside the 2D subspace we are focusing on in our analysis.  We leave a more thorough investigation of this phenomenon to future work (or potentially the camera-ready version if the reviewers think it is a high-priority item).
>
> – Regarding the input data range, we do not believe this factor has a substantial impact on our results, given that our weight initialization is random with expected mean zero (and thus the alignment of hidden unit input weights with the input data points at initialization is random, regardless of the input data range). Rerunning our Figure 1 experiments with inputs constrained to be nonnegative reproduces all the qualitative effects observed.
>
> – Regarding the point about differences between tanh / ReLU networks vanishing for more difficult tasks: we thank the reviewer for bringing up this important point.  We agree that our results in Figure 3 suggest that differences between learned representations are less prominent when the target geometry is insufficiently separable in input space relative to the input noise level.  This is also observed in Fig. 4B – the differences between activation functions vanish when input-output kernel alignment values approach 0.0, which corresponds to tasks where the output is not linearly separable in the input space.   We think it is an empirical question whether realistic tasks are in this regime or not, and we believe our (now more thorough) experiments with convolutional networks provide evidence that (at least some) realistic tasks fall in the regime where the differences between activation functions are relevant.  We believe that this is because for deep networks, the relevant notion of task difficulty becomes more complicated, as tasks in which the outputs are nonlinearly separable in input space may nevertheless be “easily separable” given the expressive capacity of a deeper network.  See our “response to all reviewers” for a discussion of new experiments we performed with multi-layer networks on synthetic tasks that explore a notion of task difficulty for deep networks.  Notably, we find tanh networks more target-aligned representations than ReLU, and this difference is in fact more pronounced for more difficult tasks.

---

> > ### Comment · Reviewer_eLPi · 2023-11-22
> > **Response to rebuttal**
> >
> > Thanks for addressing my comments! I have no further requests from the authors.

---

### Official Review · Reviewer_MMx3 · 2023-11-09

**Soundness:** 3 good
**Presentation:** 3 good
**Contribution:** 3 good
**Rating:** 8
**Confidence:** 3

**Summary:**

The manuscript analyzes how the geometry of the latent representation of a one layer neural network is influenced by the choice of the activation function, in particular ReLU and Tanh activation functions.

Using different metrics from the literature, namely kernel alignment of the latent represent, linear decodability, CCPD and sd, it is shown that the latent representation of ReLUnetworks tend to retain more information about the input, while Tanh networks align more with the output(label) representation.

Experiments are performed on multiple synthetic tasks and on Cifar 10.

**Strengths:**

*Originality*

The analysis on how the activation function can enforce a different latent representation geometry is to the best of my knowledge, novel and interesting direction to investigate.

*Clarity*

The paper is overall clear.

*Quality*

There are multiple experiments on the synthetic setting which  support quite well the claims of the paper. However, the real experiment is not strongly supported( see weaknesses section).

*Significance*

While the result on synthetic data are promising, there still some missing buts in order that make somewhat difficult to evaluate the impact of the paper ( see weaknesses section).

**Weaknesses:**

- While the analysis is interesting and it it wasn't clear to me how much it can impact can the paper have, its current form: (i) it is not clear  how much the analysis extends to real tasks: the experiments of Cifar are somewhat limited (just the alignment metric is reported and it is not clear if this behavior holds for deeper networks: i.e. some ablations on the network depth should be incorporated in the experiment in my opinion)  and the assumptions done on the synthetic tasks are unlikely to hold on larger networks (ii) there is no theory or additional experimental evidence that support why tanh and ReLU behave differently (see question section);


- The paper should report a better contextualization in the literature and comparison with similar works (it misses a related work section). I reported some works that should be discussed:

- Hayou, S., Doucet, A.; Rousseau, J.. On the Impact of the Activation function on Deep Neural Networks Training. Proceedings of the 36th International Conference on Machine Learning

- Ramachandran, Prajit, Barret Zoph, and Quoc V. Le. "Searching for activation functions." arXiv preprint arXiv:1710.05941 (2017).

- Ding, Bin, Huimin Qian, and Jun Zhou. "Activation functions and their characteristics in deep neural networks." 2018 Chinese control and decision conference (CCDC). IEEE, 2018.

- Papyan, Vardan, X. Y. Han, and David L. Donoho. "Prevalence of neural collapse during the terminal phase of deep learning training." Proceedings of the National Academy of Sciences 117, no. 40 (2020): 24652-24663.

- Zhu, Zhihui, Tianyu Ding, Jinxin Zhou, Xiao Li, Chong You, Jeremias Sulam, and Qing Qu. "A geometric analysis of neural collapse with unconstrained features." Advances in Neural Information Processing Systems 34 (2021): 29820-29834.



- Concerning clarity a better description of the metrics employed (especially SD an CCPD) would be needed, also in terms of mathematical/formal statements,if helpful.



*Minor*

I spotted a typo:

- section 5 eiegenvalues -> eigenvalues

**Questions:**

- Can the authors elaborate on the intuition of why Tanh and ReLU behave in this way? And it would be possible to derive theoretical results on this?

- The target alignment  phenomenon of Tanh relates to neural collapse [a] phenomenon : i.e. when training is kept under zero error the representation in the last layer tend to collapse in equidistance clusters aligned to the targets.
However, to the best of my knowledge this phenomenon should be agnostic of the activation function. Can the authors elaborate on this perspective?

[a] Papyan, Vardan, X. Y. Han, and David L. Donoho. "Prevalence of neural collapse during the terminal phase of deep learning training." Proceedings of the National Academy of Sciences 117, no. 40 (2020): 24652-24663.

[b] Zhu, Zhihui, Tianyu Ding, Jinxin Zhou, Xiao Li, Chong You, Jeremias Sulam, and Qing Qu. "A geometric analysis of neural collapse with unconstrained features." Advances in Neural Information Processing Systems 34 (2021): 29820-29834.


- How much the assumptions of fixing weights of the second layer is limiting in terms of measuring the four metrics employed (alignment, decodability, ccpd, sd). Is it needed just in order to approximate the dynamics?


- What is the mathematical formulation of the nonlinearity analyzed in section 7 ?

---

> ### Author Response · Authors · 2023-11-19
> **Response to reviewer MMx3**
>
> We thank the reviewer for the thoughtful comments and suggestions.  We have tried to address them as follows:
>
> – Regarding the concerns about the strength of our empirical evidence outside the synthetic task setting, see our “response to all reviewers” and Section 8 in the revised manuscript for additional results with deep convolutional networks on realistic tasks
>
> – We agree that our initial submission did not adequately address related work, especially the literature on neural collapse (which is highly related to our target alignment metric).  We have added a more comprehensive related work section, incorporating the references suggested by the reviewer and some others, to our revised manuscript.
>
> – Regarding the clarity of the description of our metrics: we have added an additional Appendix (Appendix D) that provides a clearer illustration of the definition / significance of CCGP and parallelism score.
>
> – We have included the specific form of the nonlinearities analyzed in section 7 (now section 9 in the revised manuscript), along with increased explanation of these experiments and the reasoning behind them.
>
> – Regarding intuition and theoretical explanation for the behavior of Tanh vs. ReLU: we have added a paragraph in section 4.1.1 (see red text) that provides further intuition, and additional explanation of how our approximation theoretical derivation of the input weight gradients in equation (1) gives rise to the vector fields plotted in Fig. 1D.
>
> – Regarding the question about neural collapse, it is our impression that prior work has not directly investigated the role of activation function and input data structure on neural collapse.  Prior theoretical work on the subject (such as Zhu et al’s “unconstrained features model”) uses a network parameterization that ignores the constraints imposed by the activation function / network architecture and the input data geometry, and shows that representations tend to collapse toward the label geometry under this parameterization.  Our work shows that, while this may be a good first approximation (as target alignment does tend to increase / input alignment tends to decrease over training across all networks), the specifics of network architecture and task structure play an important role in determining the extent to which neural collapse occurs. Our suite of measures of representational geometry (including CCGP, PS, and input alignment) also provides a more detailed window into the learned representational geometry.

---

> > ### Comment · Reviewer_MMx3 · 2023-11-22
> > **Response to rebuttal**
> >
> > I thank the authors for their answers and addtional experiments, and additions to the manuscript.
> > My concerns we all addressed and I updated my score accordingly.
> >
> > As an additional minor suggestion, I would suggest to merge sections 6,7,8,9 in a single structured section with paragraphs, to improve readability and visual appereance of the paper.

---

> > > ### Author Response · Authors · 2023-11-22
> > >
> > > We thank the reviewer for reading our response + revisions and updating their assessment.  We agree that merging sections 6/7/8/9 is a good idea and will do so in the camera-ready version of the paper if accepted.

---

### Author Response · Authors · 2023-11-19
**Response to all reviewers**

We thank the reviewers for their thoughtful comments on our manuscript.  We have updated our manuscript in response to reviewers’ comments.  Updates to the main text are highlighted in the revised manuscript in red text for reviewers’ convenience.  Here we highlight some points which may be of interest to all reviewers; we are also providing individual responses to reviewers.

– We have performed a more comprehensive set of experiments with deep convolutional networks on real datasets (Section 8 in the revised manuscript, Fig. 4C), including multiple architectures and multiple datasets, and reporting our full suite of metrics in all cases.  These results corroborate our main finding of higher target alignment / higher CCGP / higher parallelism score / lower input alignment when using Tanh rather than ReLU as the activation function.

– To gain a better understanding of the relevance of our analysis of shallow networks to realistic tasks / networks, we also conducted new experiments with multi-layer networks trained on synthetic tasks (Section 7 in the revised manuscript).  We focused on tasks in which the outputs were nonlinearly separable in input space, as real-world tasks require dealing with nonlinearly separable inputs.  We experimented with tasks where the input->output mapping is “maximally nonlinear” (a parity-like task in which the output depends on a nonlinear conjunction of all the binary latent variables used to generate the inputs) and tasks where the outputs were “easier” to nonlinear extract from the inputs (in particular, requiring an XOR-like nonlinear conjunction of only two of the latent variables used to generate the inputs).  We found that our main results were qualitatively corroborated in this setting: deep tanh networks learn more target-aligned representations than deep relu networks in their final layer, especially for the more difficult task.

– We have added a more comprehensive discussion of related work to the revised manuscript

– We have added further explanation for the intuition behind the analysis in Fig. 1C-D, and the intuition for why the differences between Tanh and ReLU give rise to qualitatively different dynamics in weight space.  This explanation is now also bolstered by a more comprehensive discussion in Section 9 of the role of asymptotic behavior of the activation functions vs. local behavior around the origin in explaining their impact

---

### Meta-Review · Area_Chair_MnyJ · 2023-12-06

**Metareview:**

This paper examines how the non-linearity used in ANNs alters the geometry of the learned representations. The authors show that Tanh networks usually learn representations that capture the targets, while ReLU networks learn representations that largely capture the structure of the inputs. They use analysis of the learning dynamics to pick this phenomena apart, showing how the different non-linearities and their interactions with the geometry of the inputs induce these effects.

The reviews for this paper were generally borderline-to-positive, with consistent concerns about the limited experiments, with no real-world applicable tasks, and little in the way of insights about these phenomena. However, the rebuttals addressed many of the concerns, and added new experiments in more realistic settings and more discussion of what would generate these patterns. The reviewers who responded were satisfied, and the final scores were 8,8,6,5. Given this, a decision of 'accept' was reached.

**Justification For Why Not Higher Score:**

This paper is interesting, but it's not a huge advance. It's ultimately a curiosity. Though interesting enough for a poster, I don't think it warrants a spotlight.

**Justification For Why Not Lower Score:**

The reviewers major comments were addressed, and I see no major problems with this paper.

---

### Decision · Program_Chairs · 2024-01-16

Accept (poster)